# Automatic Detection Method for Black Smoke Vehicles Considering Motion Shadows

**DOI:** 10.3390/s23198281

**Published:** 2023-10-06

**Authors:** Han Wang, Ke Chen, Yanfeng Li

**Affiliations:** 1School of Environment and Spatial Informatics, China University of Mining and Technology, Xuzhou 221116, China; ms.h.wang@cumt.edu.cn; 2College of Surveying and Land Information Engineering, Henan Polytechnic University, Jiaozuo 454000, China; 212204010020@home.hpu.edu.cn

**Keywords:** intelligent transportation, motion shadows, superpixel segmentation, YOLOv5s localization, MobilNetv3 classification

## Abstract

Various statistical data indicate that mobile source pollutants have become a significant contributor to atmospheric environmental pollution, with vehicle tailpipe emissions being the primary contributor to these mobile source pollutants. The motion shadow generated by motor vehicles bears a visual resemblance to emitted black smoke, making this study primarily focused on the interference of motion shadows in the detection of black smoke vehicles. Initially, the YOLOv5s model is used to locate moving objects, including motor vehicles, motion shadows, and black smoke emissions. The extracted images of these moving objects are then processed using simple linear iterative clustering to obtain superpixel images of the three categories for model training. Finally, these superpixel images are fed into a lightweight MobileNetv3 network to build a black smoke vehicle detection model for recognition and classification. This study breaks away from the traditional approach of “detection first, then removal” to overcome shadow interference and instead employs a “segmentation-classification” approach, ingeniously addressing the coexistence of motion shadows and black smoke emissions. Experimental results show that the Y-MobileNetv3 model, which takes motion shadows into account, achieves an accuracy rate of 95.17%, a 4.73% improvement compared with the N-MobileNetv3 model (which does not consider motion shadows). Moreover, the average single-image inference time is only 7.3 ms. The superpixel segmentation algorithm effectively clusters similar pixels, facilitating the detection of trace amounts of black smoke emissions from motor vehicles. The Y-MobileNetv3 model not only improves the accuracy of black smoke vehicle recognition but also meets the real-time detection requirements.

## 1. Introduction

Traditional control of motor vehicle exhaust pollution mainly occurs during processes such as vehicle registration and annual inspections rather than effective supervision during vehicle usage. The application of onboard detection technology and road remote sensing monitoring technology can identify motor vehicles emitting black smoke exhaust on roads. However, the size of detection devices is relatively large, making it difficult to deploy them extensively on urban roads. In recent years, with the rapid development of artificial intelligence, methods for automatically detecting black smoke-emitting vehicles based on monitoring videos from road surveillance cameras have become more intelligent and efficient. Cao et al. [1] utilized the Inceptionv3 convolutional neural network to capture spatial information of suspected black smoke frames in monitoring videos, while a long short-term memory network learned the temporal dependencies between video frames. They built a dual-branch black smoke vehicle detection network based on the CenterNet [2] framework, utilizing vehicle feature maps to generate attention mechanisms for guiding the training of black smoke feature maps. This model achieved a detection speed of 25.46 FPS and mAP@0.5 of 92.5%. Xia et al. [3] proposed using a convolutional neural network model based on LeNet-5 to detect vehicles emitting black smoke. Simultaneously, an Inception module was introduced, and multiple convolutional kernels of different sizes were used to perform convolution operations to extract black smoke features. Zhang et al. [4] proposed a multi-frame classification network based on 2D-3D fusion for detecting black smoke-emitting vehicles. They utilized both 2D and 3D convolutions to extract spatial and spatiotemporal features of black smoke. The model achieved a recognition accuracy of 90.3%, with an average inference time of 45.9 ms per frame. Zhang et al. [5] designed two lightweight networks, YOLOv3-M3-CBAM and YOLOv4-GhostNet, based on the YOLOv3 and YOLOv4 models. After improvement, both models achieved a detection speed of 20 FPS. Liu and others proposed a black smoke vehicle detection model based on a three-dimensional convolutional network and a non-local attention mechanism. This model utilizes three-dimensional convolutional kernels to learn the spatial features and temporal information of black smoke videos. It jointly evaluates the existence of black smoke by considering suspected black smoke regions across multiple consecutive frames [6].

The aforementioned automatic detection methods for vehicles emitting black smoke primarily focus on improving and optimizing model structures based on the target features of black smoke emissions. However, factors that interfere with black smoke vehicle detection in real-world scenarios have not been taken into consideration. For instance, when vehicles are driving under clear weather conditions, they cast dynamic shadows. These dynamic shadows exhibit certain visual similarities to black smoke emissions, which significantly affect the recognition accuracy of black smoke vehicle detection. In areas where shadows are cast and exhibit high brightness and saturation, their color values closely follow a linear relationship with the background image. This principle can be employed for shadow detection, where the brightness in shadow areas is lower than that in non-shadow areas, while chromaticity remains consistent [7]. Khan et al. [8] employed multiple supervised convolutional deep neural networks to learn shadow-related features. However, due to a lack of labeled training data, this approach remains challenging in practical application scenarios. Tian et al. proposed a normalized cross-correlation method based on texture features, which involves calculating the NCC value by comparing the texture similarity between the current frame and the background pixels at the same position and their neighboring pixels for shadow judgment [9]. Shadow removal involves restoring shadow regions in an image while preserving attributes such as texture and color on the object’s surface. Shadow binary masks and shadow masks are commonly used for conditional information for generators in generative adversarial networks. Shadow binary masks often utilize alpha matting techniques to label shadow and non-shadow regions, but shadow masks can be easily influenced by human errors [10,11]. The challenge in shadow detection lies in accurately identifying the shadowed areas on object surfaces, while the challenge in shadow removal is to protect object surface information from being altered. However, due to the certain similarity between black smoke emissions and dynamic shadows, the solution of detecting and then removing dynamic shadows is difficult to implement in the task of automatic detection of vehicles emitting black smoke.

The existing intelligent algorithm for detecting smoky vehicles faces several challenges, including difficulties in model deployment, limited model applicability, and the need to improve accuracy in smoky vehicle identification. The large number of model parameters and computational requirements make model deployment challenging, necessitating the development of a more lightweight smoky vehicle detection network. The limited model applicability and low recognition accuracy are due to the fact that existing methods have not adequately considered factors that interfere with the smoky vehicle detection process during optimization and improvement, such as the motion shadows produced by motor vehicles on sunny days. Therefore, this study has designed an automatic smoky vehicle detection solution that takes into account motion shadows, as shown in Figure 1. Based on the “segmentation-classification” concept, it cleverly addresses situations where motion shadows coexist with smoky exhaust, and it achieves this by using a superpixel segmentation algorithm called simple linear iterative clustering to cluster and re-segment similar pixels in the image [12]. Directly detecting smoky exhaust using YOLO series object detection models faces challenges such as missing small targets, misidentifying motion shadows, and difficulty in associating detected smoky exhaust with motor vehicles in high-traffic areas [13]. However, by locating moving objects that include motor vehicles, smoky exhaust, and motion shadows, the target positioning effect is superior to traditional motion object detection methods. This approach can exclude irrelevant moving objects, such as roadside trees, that are not related to the research being conducted. The images of moving objects are processed using the superpixel segmentation algorithm to obtain superpixel images belonging to three categories: motor vehicles, smoky exhaust, and motion shadows, which serve as training samples. The design of a lightweight network structure, compared with convolutional neural networks, is more suitable for real-time detection tasks. Therefore, the obtained segmented samples of different categories are fed into the smoky vehicle automatic detection model built on the lightweight MobileNetv3 network [14,15,16] for recognition and classification. In the task of automatic smoky vehicle detection, not only accurate identification of smoky vehicles is required, but also the network inference speed needs to be improved, especially when dealing with a large amount of surveillance video data.

## 2. Locating Moving Objects

### 2.1. Object Detection Model

Object detection, as a fundamental problem in computer vision research, involves precisely locating all objects of given classes in an image and predicting the class for each object. The traditional object detection process can be roughly divided into three steps: candidate box generation, feature vector extraction, and region classification. Deep learning-based object detection methods allow for end-to-end learning, eliminating the need for staged training during the process. These methods include two-stage detection algorithms based on candidate windows and single-stage detection algorithms based on regression. Single-stage detection algorithms do not require generating candidate regions and can directly predict the class probabilities and location information of objects. The YOLO series of algorithms improve accuracy through end-to-end training, and they are compatible and suitable for industrial applications [17,18]. In 2020, YOLOv5 was introduced, followed by the YOLOX model proposed by Megvii in the following year. In 2023, Ultralytics continued to upgrade and optimize the previously introduced YOLOv5 model and released the YOLOv8 model. The performance comparison of these three different models is shown in Table 1. The YOLOXs model and the YOLOv5s model both use Focus and CSPDarknet53 as the backbone networks, and the neck network adopts the FPN + PAN structure. Activation functions include LeakyReLU and Sigmoid, with LeakyReLU used in the hidden layers and Sigmoid used in the detection layers. The YOLOXs model uses a free anchor box strategy for the prediction layer, while the YOLOv5 model learns anchor boxes automatically from the training dataset, reducing the original three anchor box candidates to one and directly predicting the four parameters for each target box [19]. The main feature of YOLOv8 is its scalability, which can be applied not only to YOLO series models but also to non-YOLO models and tasks such as segmentation, classification, and pose estimation. There have been significant improvements in the neck part of the network, where all C3 modules have been replaced with C2f modules, and all CBS modules before upsampling have been removed, with upsampling operations directly performed using C2f modules [20]. YOLOv5 uses a simple convolutional neural network architecture, while YOLOv8 employs multiple residual units and branches and is more complex. Table 1 presents the test results comparison of different object detection models on our custom dataset in this study. YOLOv5 has a smaller parameter count, faster inference speed, and is more suitable for real-time motor vehicle detection.

The overall structure of the YOLOv5s model consists of an input layer, backbone network, neck network, and prediction layer; as shown in Figure 2. Image preprocessing includes mosaic data augmentation, adaptive image scaling, and adaptive anchor boxes. Mosaic data augmentation involves combining four images through random cropping, flipping, and other methods. This enhances the network’s robustness and addresses issues of insufficient dataset samples and uneven size distribution [21]. The backbone feature extraction network is primarily composed of Conv modules, C3 modules, and SPPF modules. In version 6.0, the previous version’s focus module has been replaced with a convolutional layer with a kernel size of 6, stride of 2, and padding of 2. For GPUs with limited performance, using a convolutional layer in this context is more efficient than using the focus module. While earlier versions used the CSP module to reduce model computation and achieve cross-layer fusion of local image features, version 6.0 employs the C3 module with a similar role. The difference lies in the removal of the Conv after concatenation, and the standard convolution module after Concat has replaced the Relu activation function with SiLU. In version 6.0, the SPP module is replaced with the SPPF module, both of which aim to fuse output features and enlarge the object receptive field [22,23]. The neck network combines a feature pyramid network with a path aggregation network to reprocess features extracted at different stages. The feature pyramid network transfers strong semantic information from deep feature maps to shallow ones through upsampling, while the path aggregation network transfers positional information from shallow feature maps to deep ones through downsampling. This simultaneous upsampling and downsampling achieves multi-scale feature fusion [24,25]. The prediction layer is responsible for detecting the class and position of target objects. It mainly consists of the loss function and non-maximum suppression. The loss function is the sum of localization loss, confidence loss, and classification loss. Non-maximum suppression is employed to eliminate redundant bounding boxes.

### 2.2. Motion Object Extraction

Motion object detection is a crucial component of intelligent video surveillance systems. Currently, mainstream methods for motion object detection include optical flow, frame differencing, and background subtraction [26,27,28]. Background subtraction involves comparing the current image with a background image. This method can adapt to changes in application scenarios and handle noise disturbances to some extent [29,30]. Frame differencing is simple to implement, has low computational requirements, and exhibits strong adaptability and robustness in dynamic environments. However, in the presence of large areas of similar grayscale values on the surface of the moving object, frame differencing may result in holes in the image [31,32]. In recent years, deep learning technology has shown its remarkable feature extraction capabilities. Object detection algorithms can locate motion objects, thereby predefining the scope of study and reducing the interference of influencing factors. Two-stage object detection algorithms have slow processing speeds, making them inadequate for real-time detection tasks. On the other hand, the YOLO series of one-stage object detection algorithms can significantly improve detection speed while sacrificing only a slight decrease in accuracy. Thus, this study chooses the YOLOv5 model, which excels in object detection performance, to locate the regions of moving objects in road traffic surveillance videos. Based on network depth and width, the model is available in four sizes: small, medium, large, and extra-large. In practical applications, there is a need to balance the relationship between model accuracy, speed, and volume. Considering the relatively small dataset and the requirement for real-time detection, the YOLOv5s model with the smallest volume is selected to locate motion objects. The extracted motion object regions include moving vehicles, black smoke emissions from the tailpipes, and the dynamic shadows generated by vehicles under clear weather conditions. Figure 3 demonstrates the motion object regions with both black smoke emissions and dynamic shadows extracted by the YOLOv5s model from road traffic surveillance videos.

## 3. Motion Target Segmentation

### 3.1. Optimal Segmentation Parameters

Image segmentation involves dividing an image into different regions with specific semantic meanings based on certain similarity criteria. In the early days, image segmentation was mostly performed at the pixel level, using a two-dimensional matrix to represent an image, without considering the spatial relationships between pixels [33]. Simple linear iterative clustering uses the similarity of features between pixels to group pixels and classify pixels of the same type. This is advantageous for reducing data dimensions and computational complexity, thus enhancing the efficiency of image processing [34]. The objective of this research is to automatically detect vehicles emitting black smoke emissions. However, the presence of dynamic shadows generated by vehicles under clear weather conditions can impact the accuracy of black smoke vehicle detection. Therefore, a superpixel segmentation algorithm is employed to process the images of the regions, with moving objects extracted by the YOLOv5s model. This process aims to obtain superpixel images belonging to three categories: vehicles, black smoke emissions, and dynamic shadows. These superpixel images are then used as training samples.

The implementation process of the SLIC involves converting a color image into a five-dimensional feature vector V = [L, a, b, x, y] in the CIELAB color space and XY coordinates. Each pixel’s color vector (LL, aa, bb) and position vector (xx, yy) together form a five-dimensional feature vector, enabling the local clustering of image pixels [35]. Firstly, the color space conversion is performed, and a nonlinear tone mapping of the image is achieved using the gamma function. The initial set of k superpixel seed points is evenly distributed over the image containing N pixels [36]. The generated seed points might fall on the edges of superpixels with significant gradients or noisy pixel locations. Therefore, the initial seed points are generally chosen as the positions with the smallest gradient values within a 3 × 3 neighborhood. The similarity between pixel points and seed points is measured using a distance metric that combines color distance and spatial distance. The parameter m represents a weight factor that gauges the relative importance between color and spatial distances, while S denotes the distance between adjacent seed points. The value of D indicates the similarity between two pixels, with higher values implying greater similarity [37].
(1)$$d_{Lab}=\sqrt{{\left(L_{i}-L_{j}\right)}^{2}+{\left(a_{i}-a_{j}\right)}^{2}+{\left(b_{i}-b_{j}\right)}^{2}}$$
(2)$$d_{xy}=\sqrt{{\left(x_{i}-x_{j}\right)}^{2}+{\left(y_{i}-y_{j}\right)}^{2}}$$
(3)$$D=\sqrt{{\left(d_{Lab}\right)}^{2}+{\left(\frac{d_{xy}}{S}\right)}^{2}m^{2}}$$

In the equation, $L_{i}$, $a_{i}$, and $b_{i}$ represent the three channel components of a pixel in the CIELAB color space, while $x_{i}$ and $y_{i}$, respectively, denote the horizontal and vertical coordinates of pixel i.

To enhance the computational efficiency of the SLIC, a search for similar pixels is conducted within a 2S × 2S region centered around the seed point. Clustering involves calculating the distance metric between all pixels within this region and the seed point. Through repetitive iterations and assignments, similar feature pixels are grouped to form super pixel blocks. The initial number of seed points, k, and the weight factor, m (which determines the relative importance between color distance and spatial distance), both influence the generation of the superpixel image [38,39]. Therefore, in this experiment, a controlled variable method is employed to analyze and compare the effects of different parameter combinations on the segmentation of motion object regions. This analysis aims to determine the optimal parameter values for the SLIC.

In the first set of comparative experiments, the balancing parameter m of the SLIC was set to 10 and the number of seed points, *k*, was set to 500, 1000, 1500, and 2000, respectively. The segmentation results of motion object regions are shown in Figure 4. In Figure 4, the red rectangular boxes highlight the segmentation outcomes at the junctions between vehicle tail, dynamic shadow, and road surface. As the number of seed points increases, the under-segmentation phenomenon at the junctions of different objects gradually diminishes, resulting in more consistent content within the generated superpixel blocks. When the segmentation accurately captures the junctions between different objects, increasing the number of seed points will lead to a higher number of superpixel blocks generated during motion object region segmentation. Consequently, this can amplify the computational workload during model classification. Considering the segmentation outcomes from the four different parameter settings, the best segmentation results were achieved when the number of seed points, k, was set to 1500.

The second set of comparative analysis experiments involved setting the number of seed points in the SLIC to 1500. The balancing parameter was varied as 5, 10, 15, and 20, respectively. The segmentation results of motion object regions are shown in Figure 5. In Figure 5, the red rectangular boxes highlight the segmentation details at the junction between black smoke emissions and the road surface. When the balancing parameter is set too small, the boundaries of the object’s contours appear blurry. Conversely, when the balancing parameter is set too large, the boundary segmentation of the object’s contours becomes imprecise. Considering the segmentation outcomes from the four different parameter settings, the best segmentation results were achieved when the balancing parameter m was set to 10. Consequently, the optimal parameters for the SLIC in this application scenario are selected as k = 1500 and m = 10.

### 3.2. Creating Dataset

The three essential elements of deep learning are data, algorithms, and computing power. Data hold a crucial position in deep learning, as a high-quality dataset often improves the accuracy of model predictions. When data are scarce, it is also crucial to utilize existing data resources to create high-quality datasets. A high-quality dataset not only considers the quantity and quality of the raw data but also takes into account the factors that can interfere with experiments during the data preprocessing process. In this study, the data are sourced from road traffic monitoring videos, and the research goal is to automatically detect motor vehicles emitting black smoke on the road. First, the original images containing motor vehicles are obtained through video frame-by-frame processing and selection, as shown in Figure 6.

Based on the YOLOv5s model, we located moving targets and obtained a total of 2900 images containing motor vehicles. Next, based on the 2900 images of located moving targets, two sets of experimental plans were designed to obtain training samples for different models. The automatic detection model for black smoke vehicles considering motion shadows is referred to as “Y-MobileNetv3”, while the model not considering motion shadows is referred to as “N-MobileNetv3”. The extracted images of moving targets were processed using a superpixel segmentation algorithm, resulting in 1082 images of black smoke emissions, 1035 images of motion shadows, and 1118 images of motor vehicles as training samples for the Y-MobileNetv3 model. The extracted images of moving targets include heavy-duty trucks, medium-sized vans, and light sedans. Adaptive thresholds were designed based on the aspect ratios of the extracted images of moving targets. The last third of the images was selected as the suspected black smoke region, resulting in a total of 2320 non-black smoke emissions and 580 black smoke emissions used as input for training the N-MobileNetv3 model. The process for creating training samples with and without considering motion shadows is shown in Figure 7. The experimental process ensures the consistency of YOLOv5s in locating images of moving targets, with the difference being that the training samples for the model considering motion shadows undergo superpixel segmentation to classify non-black smoke emissions into motor vehicles and motion shadows as two separate categories.

The settings of two key parameters in the superpixel segmentation algorithm need to be adjusted according to the specific application scenarios. When selecting training samples from different categories after motion target segmentation, it is important to ensure that superpixel images taken from the center of each category region are preserved. This approach helps avoid issues related to excessive segmentation of neighboring objects from different categories, which can negatively impact the quality of training samples. Superpixel images with a resolution of 100 × 100 are saved, as shown in Figure 8, for training samples of some motor vehicles, black smoke emissions, and motion shadows. For motor vehicles, key features that are easy to identify, such as vehicle taillights, rear bumpers, and vehicle body colors, are selected for the superpixel images. The dataset covers various types of motor vehicles, including heavy-duty trucks, medium-sized vans, and light sedans. Superpixel images of black smoke emissions exhibit a hazy and blurry appearance with no distinct texture features, while superpixel images of motion shadows have clearer texture features. These visual differences help distinguish between the two categories.

## 4. MobileNetv3 Classification

In 2017, the Google team introduced the lightweight MobileNetv1 model. While ensuring model accuracy, this model significantly reduced the computational load of network model parameters, making it suitable for running applications on mobile terminal devices. Compared with the traditional convolutional neural network VGG16 model, the MobileNetv1 model had 1/32 of the parameters, while only sacrificing 0.9% of classification accuracy [40,41]. The MobileNetv2 model is an optimized and upgraded version of the MobileNetv1 model by the Google team. It boasts higher accuracy and a smaller model size. This model dramatically reduces the computational load of parameters, making it highly efficient for deployment on mobile devices and suitable for real-world applications. Similar to MobileNetv1, the design of the MobileNetv2 model’s architecture also incorporates depthwise separable convolutions instead of standard convolutions. A pointwise convolution is added before the depthwise convolution to increase the dimensionality, allowing the network model to extract features in a higher-dimensional space [42]. Drawing inspiration from the design philosophy of the ResNet network architecture, the input and output are added together in the model, facilitating the flow of information between layers; this aids in feature reuse during forward propagation and mitigates the vanishing gradient problem during backward propagation. The most innovative aspect of the MobileNetv2 model’s architecture design is the inverted residual structure. A shortcut connection is only established when the stride is 1 and the input and output feature matrices have the same shape.

The inverted residual structure shown in Figure 9 utilizes a 1 × 1 pointwise convolution before the depthwise separable convolution to increase the channel dimension of the feature map, followed by a 1 × 1 convolution for dimension reduction. The classic order of residual blocks is reversed to form the inverted residual structure. The ReLU6 activation function is employed within the inverted residual structure, while the linear activation function is used in the final 1 × 1 convolution layer. In this context, using the ReLU6 activation function would lead to significant loss of low-dimensional feature information. The overall design of the inverted residual structure is characterized by narrower channels at the two ends and a wider middle section. Applying a linear activation function helps mitigate information loss in the output. The MobileNetv3 model, proposed by Howard and his team in 2019, continues to utilize depthwise separable convolutions from the v1 version and the inverted residual structure from the v2 version [43]. The MobileNetv3 model introduces a new SE (squeeze and excitation) attention mechanism and replaces the swish activation function with the h−swish activation function. The SE attention mechanism comprises compression and excitation parts, involving two fully connected layers with Relu6 and h−swish activation functions, respectively, after global average pooling of features [44,45]. The original authors approximated the swish activation function with ReLU6 to create the h−swish activation function, which effectively addresses the issue of complex gradient calculation [46,47]. The computation formula for the h-swish activation function is as follows:(4)swish(x)=x⋅sigmoid(βx)
(5)Relu=max(0,x)
(6)$$h-swish(x)=x\frac{Relu(x+3)}{6}$$

In the equation, x represents the input and β is a constant or a training parameter.

The MobileNetv3 model strengthens feature extraction through a combination of 3 × 3 standard convolutions and the neck structure. It further enhances the model by incorporating a max pooling layer, substituting 1 × 1 convolution blocks for fully connected layers, and implementing a series of operations to reduce network parameters and complexity [48]. The MobileNetv3 model comes in two scale sizes: “large” and “small”. In the ImageNet classification competition, the MobileNetv3-large network achieved a 4.6% increase in accuracy and a 5% improvement in detection speed compared with the v2 version [49]. Similarly, the MobileNetv3-small network demonstrated a 3.2% accuracy improvement and a 15% increase in detection speed over the v2 version.

Taking into account the small size of the experimental dataset and the real-time detection requirements, the MobileNetv3-small model, which has a smaller volume, was chosen for identifying black smoke-emitting vehicles in this study. The training process of the Y-MobileNetv3 model for automatic detection of black smoke-emitting vehicles with consideration of motion shadows is depicted by the loss function variation curve in Figure 10. As the training epochs reach 120 rounds, the loss function fluctuates between 0.1 and 0.2, indicating that the model training is effective and stable.

## 5. Experimental Results and Analysis

### 5.1. Experimental Environment Configuration

The experimental hardware and software environment configuration parameters are shown in Table 2. The hyperparameters of the YOLOv5s model were determined based on previous relevant research and comparative experiments, with input image resolution uniformly scaled to 640 × 640. Prior to training, the initial anchor boxes were clustered using the k-means algorithm, resulting in (10, 13, 16, 30, 33, 23), (30, 61, 62, 45, 59, 119), and (116, 90, 156, 198, 373, 326) The YOLOv5s model was trained for a total of 200 epochs, with a batch size of 8. The Adam optimizer was selected, and the initial learning rate was set to 1 × 10^−3^ with an initial decay rate of 1 × 10^−5^. The learning rate reduction was performed using the cosine annealing strategy. For the MobileNetv3 model, the initial learning rate was set to 0.0001, and the batch size was set to 16 for a total of 130 epochs. The training process utilized the mosaic data augmentation method to enhance the model’s robustness, and the SGD optimizer was employed for gradient updates during training.

### 5.2. Comparative Experimental Analysis

The test results for automatic detection of black smoke vehicles based on the MobileNetv3 model are shown in Table 3. Y-MobileNetv3 represents the automatic detection model for black smoke vehicles considering motion shadows. The training samples input for Y-MobileNetv3 are superpixel images obtained through segmentation of motion target regions extracted by YOLOv5s. N-MobileNetv3 represents the automatic detection model for black smoke vehicles without considering motion shadows. The training samples input for N-MobileNetv3 are motion target images extracted by YOLOv5s. The confusion matrix, also known as an error matrix, is capable of determining the quality of the model’s classification. Predicted values and actual values for all classes are placed in the same table, providing a clear view of the number of correct and incorrect recognitions for each class. Each column of the confusion matrix represents the predicted class of images, with the values indicating the number of images predicted for each class. Each row of the confusion matrix represents the actual class of images, with the values indicating the number of images belonging to each actual class. The results of the confusion matrix can be used to calculate more advanced classification evaluation metrics such as average accuracy, precision, and recall. Average accuracy is the most commonly used classification evaluation metric, calculated by dividing the number of correctly classified instances by the total number of samples. A higher value indicates better classification performance of the model.

The average accuracy variation curves based on the MobileNetv3 model are presented in Figure 11. The red curve represents the Y-MobileNetv3 model for automatic detection of black smoke vehicles considering motion shadows, while the black curve represents the N-MobileNetv3 model for automatic detection of black smoke vehicles without considering motion shadows. Observing the average accuracy variation curves reveals that the trends of average accuracy for both models change similarly with the epochs, and their learning efficiency is comparable. When the training epochs reach around 80, the average accuracy of the Y-MobileNetv3 model fluctuates around 95%, while the average accuracy of the N-MobileNetv3 model fluctuates around 90%.

Through the confusion matrix in Table 3, we can compute the model evaluation metrics, as shown in Table 4. The average accuracy of the Y-MobileNetv3 model is 95.17%, while the average accuracy of the N-MobileNetv3 model is only 90.34%. Average accuracy is an evaluation metric for the entire classification model, but for evaluating each category, we primarily use precision and recall. Precision refers to the proportion of samples identified by the model as black smoke exhaust that are actually black smoke exhaust. Recall is the proportion of actual black smoke exhaust samples that the model correctly predicts as black smoke exhaust. The Y-MobileNetv3 model has a precision of 96.03% and a recall of 94.77%, both of which are 4.64% and 4.58% higher than the N-MobileNetv3 model, respectively. The Y-MobileNetv3 model has a single-image inference speed of 7.3 ms, slightly faster than the N-MobileNetv3 model. This improvement is due to the superpixel segmentation algorithm that groups and classifies similar pixels, enhancing the efficiency of model recognition and classification computations. Compared with existing research on black smoke vehicle detection algorithms, the algorithm proposed in this study, which takes into account motion shadows, has advantages in both detection speed and accuracy, as shown in Table 5. The most important contribution of this research is that it goes beyond previous detection algorithms that solely rely on improving the model network structure to enhance detection performance. Instead, it considers the mutual influence between the research objectives and interfering factors, thereby improving both recognition accuracy and model generality. Under the same test dataset, the Y-MobileNetv3 model’s average accuracy improves by 4.73%, clearly demonstrating that using the superpixel segmentation algorithm in the data preprocessing phase to process motion target images and classify motion shadows as a separate category can effectively enhance the recognition accuracy and computational efficiency of the automatic black smoke vehicle detection model.

The results of the Y-MobileNetv3 model are illustrated in Figure 12. Figure 12a, depicts an example where the moving object consists solely of black smoke exhaust. In Figure 12b, an example shows a moving object consisting exclusively of motion shadows. In Figure 12c, an instance demonstrates the coexistence of black smoke exhaust and motion shadows. On clear days, motor vehicles generate motion shadows, and the Y-MobileNetv3 model is capable of excluding the interference of motion shadows and accurately identifying black smoke exhaust. The left side of Figure 9 displays the motion object regions extracted by the YOLOv5s model, while the right side showcases the visualized images of the Y-MobileNetv3 model’s test results. Superpixels marked in green represent black smoke exhaust, while those in red denote motion shadows. The motion object regions are recognized and classified by the Y-MobileNetv3 model. The presence of superpixel images indicating black smoke exhaust in the classification results serves as the basis for determining whether a motor vehicle emits black smoke. When black smoke exhaust and motion shadows coexist within the same superpixel block, the model’s classification will identify it as black smoke exhaust.

The YOLOv5s model locates the moving target regions of motor vehicles, effectively avoiding interference from other irrelevant moving objects in the research. Experimental results indicate that the N-MobileNetv3 model exhibits false positives and false negatives when detecting motor vehicles emitting trace amounts of black smoke exhaust. In contrast, the Y-MobileNetv3 model can accurately identify them. As shown in Figure 13, the primary reason for false positives and false negatives in the N-MobileNetv3 model is the imprecise identification of suspected black smoke regions. However, the Y-MobileNetv3 model identifies the entire motion target region obtained through the superpixel segmentation algorithm, allowing for accurate recognition of motor vehicles emitting trace amounts of black smoke exhaust. The superpixel segmentation algorithm groups pixels based on the similarity of their features. This characteristic not only aids in distinguishing between black smoke exhaust and motion shadows but also assists the model in identifying motor vehicles emitting trace amounts of black smoke exhaust. By processing the extracted motion target regions using the superpixel segmentation algorithm and classifying motion shadows as a separate category, it effectively improves the recognition accuracy of automatic black smoke vehicle detection.

## 6. Conclusions

In the context of road traffic surveillance videos, deep learning-based methods can be employed for automatic detection of black smoke-emitting vehicles. However, these methods often suffer from challenges such as lower recognition accuracy and limited model generalization. The “segmentation-classification” approach effectively distinguishes between black smoke exhaust and motion shadows, reducing instances where motion shadows are misclassified as black smoke exhaust. This approach breaks away from the conventional technique of detecting first and then removing shadows, enhancing both the accuracy of identifying black smoke-emitting vehicles and the general applicability of the automatic detection model. Using the same test dataset, the Y-MobileNetv3 model for black smoke vehicle automatic detection, which considers motion shadows, achieves an average accuracy of 95.17%, precision of 96.03%, and recall of 94.77%. In comparison with the N-MobileNetv3 model, which does not consider motion shadows, all evaluation metrics show significant improvement in results, and the Y-MobileNetv3 model also demonstrates faster inference speeds. The recognition computation time for the Y-MobileNetv3 model is 7.3 ms per image, ensuring real-time detection of black smoke-emitting vehicles while maintaining accuracy.

The model’s recognition and classification results are visually displayed through color-coded superpixel images, effectively illustrating the model’s successful differentiation between black smoke exhaust and motion shadows. The SLIC aggregates and classifies neighboring pixels with similar features, not only distinguishing between black smoke exhaust and motion shadows but also significantly enhancing the model’s deployment applicability. The superpixel images generated during image segmentation are beneficial for detecting vehicles emitting small amounts of black smoke exhaust, thereby reducing the false negative rate of the automatic detection model. Currently, quantitative calculation of black smoke exhaust concentration from road surveillance video data using computer vision technology remains challenging. However, a color-coded approach can roughly depict the outline of black smoke exhaust. Further research will be to conduct in-depth research on image segmentation of moving targets, to further explore the differences between black smoke exhaust and moving shadows, with the aim of more accurately depicting the black smoke exhaust outline. It realizes the hierarchical classification management of smoky vehicles and helps relevant law enforcement departments to efficiently monitor smoky vehicles.

## Figures and Tables

**Figure 1 sensors-23-08281-f001:**
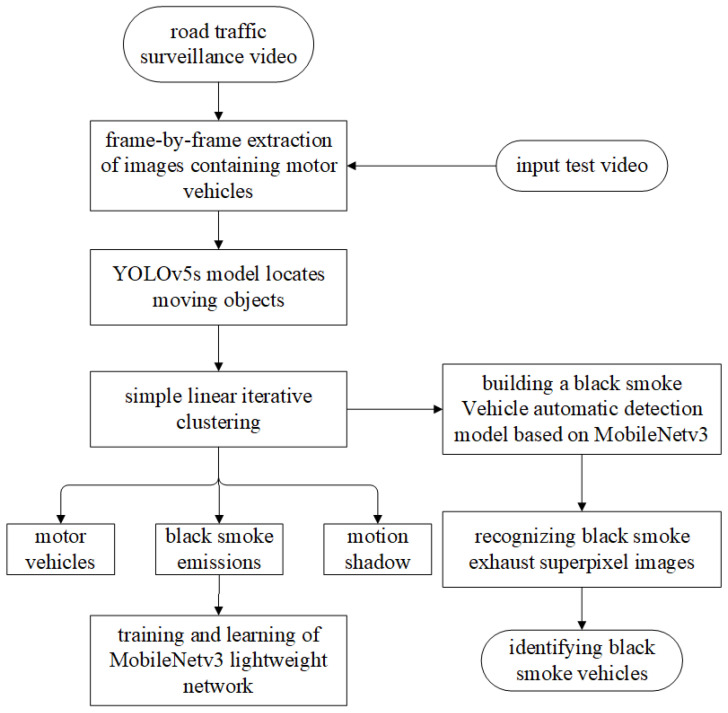
The flowchart for smoky vehicle detection considering motion shadows.

**Figure 2 sensors-23-08281-f002:**
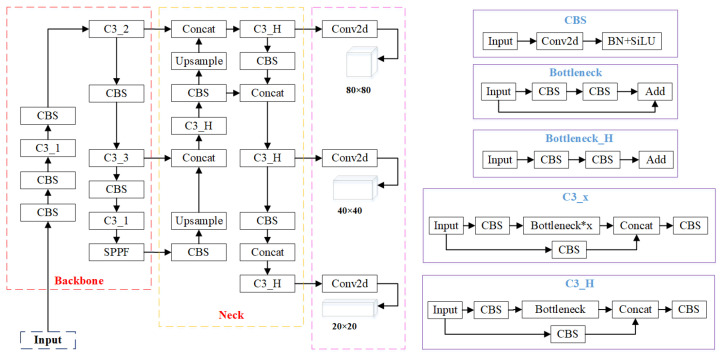
Overall architecture of YOLOv5s model.

**Figure 3 sensors-23-08281-f003:**
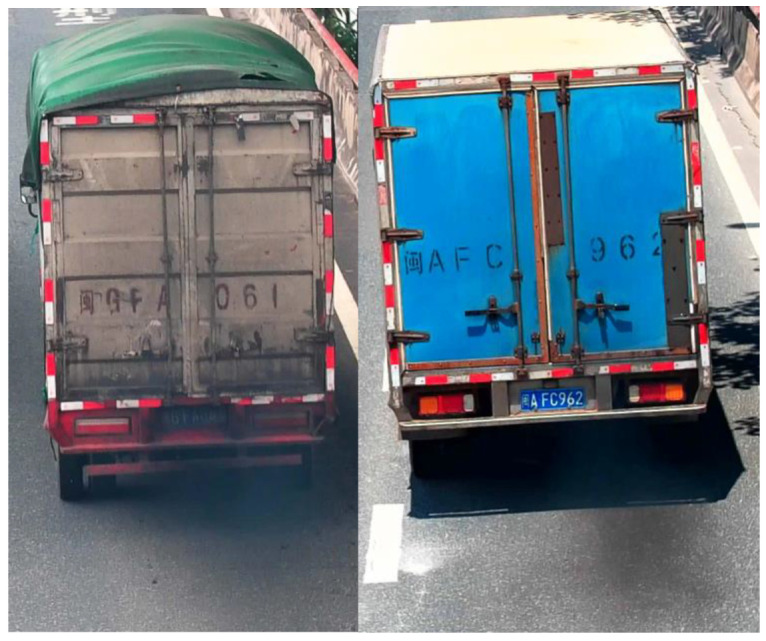
YOLOv5s model to extract motion target regions.

**Figure 4 sensors-23-08281-f004:**
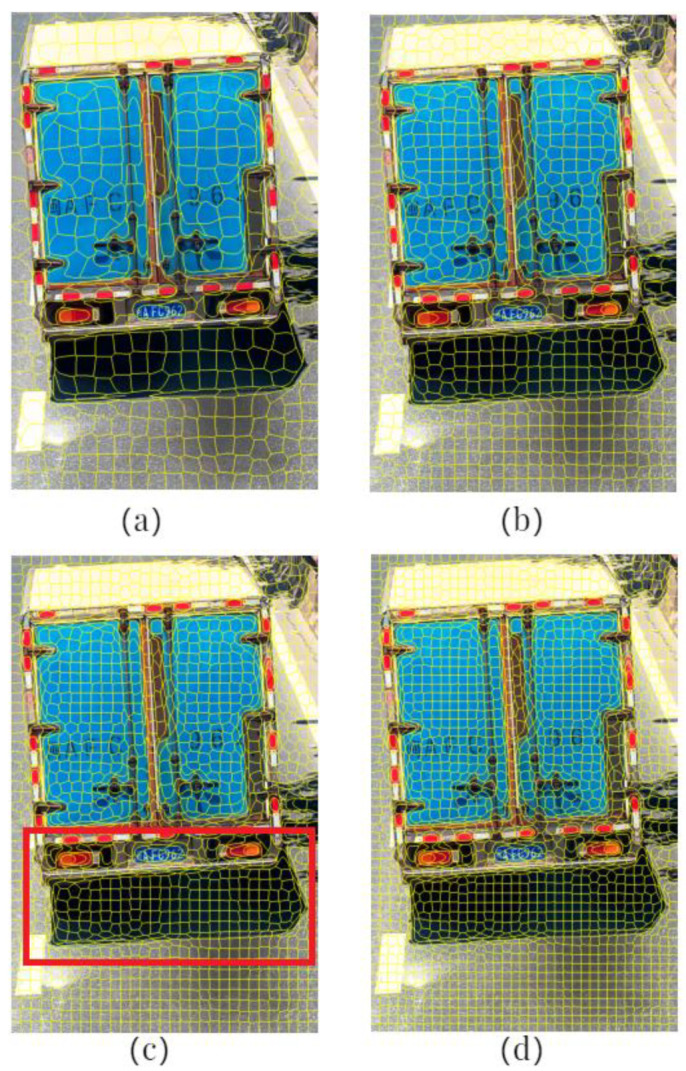
Segmentation results with different numbers of seed points when m = 10: (**a**) *k* = 500; (**b**) *k* = 1000; (**c**) *k* = 1500; (**d**) *k* = 2000.

**Figure 5 sensors-23-08281-f005:**
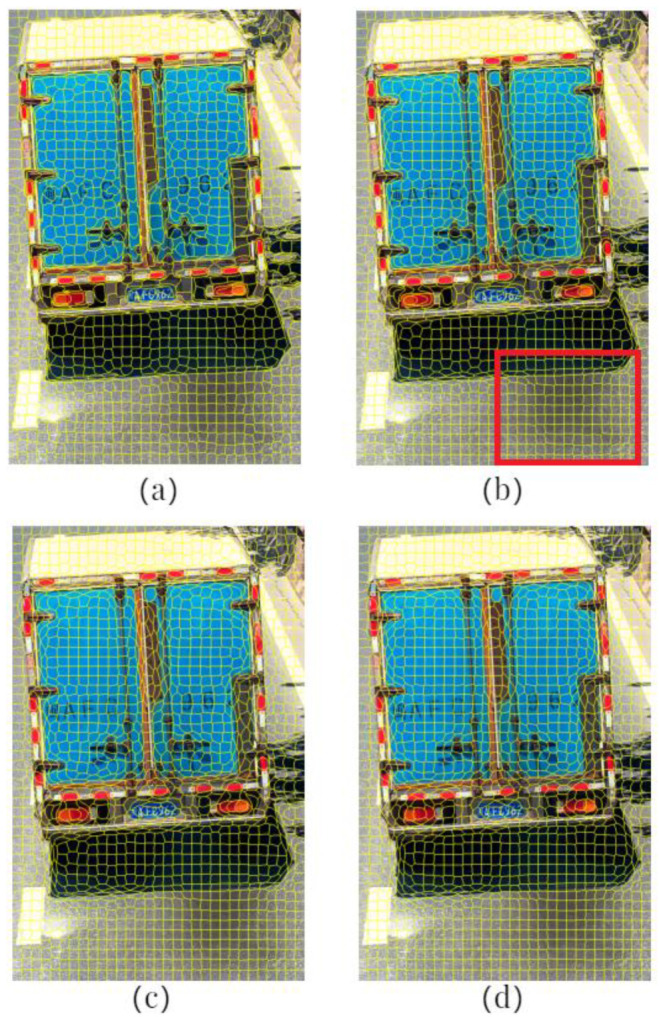
Segmentation results with different balance parameters when k = 1500: (**a**) *m* = 5; (**b**) *m* = 10; (**c**) *m* = 15; (**d**) *m* = 20.

**Figure 6 sensors-23-08281-f006:**
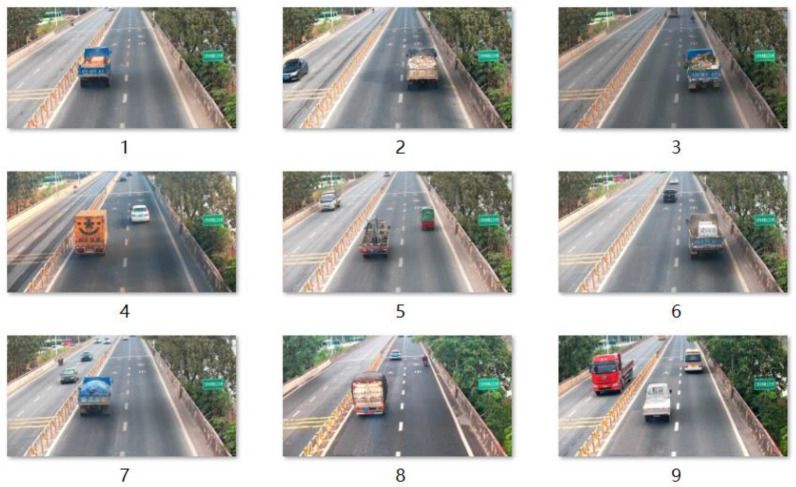
Video frame-by-frame processing to obtain original images containing motor vehicles: 1–3 and 6–7 are heavy truck; 4 and 8 are medium truck; 5 and 9 are light truck.

**Figure 7 sensors-23-08281-f007:**
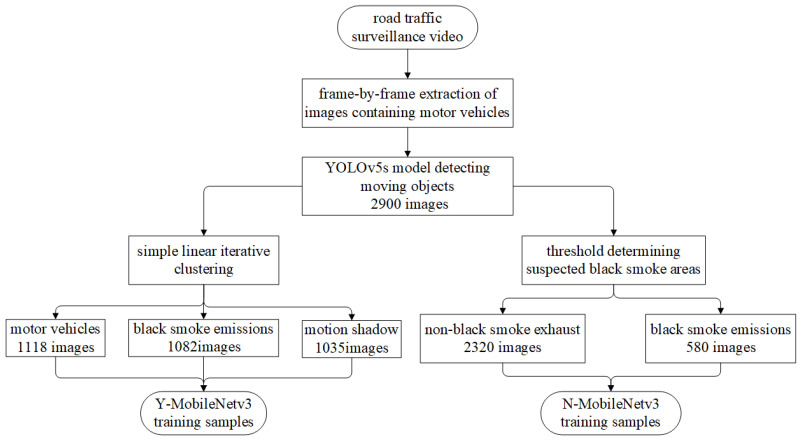
The flowchart for creating training samples for Y-MobileNetv3 and N-MobileNetv3 model.

**Figure 8 sensors-23-08281-f008:**
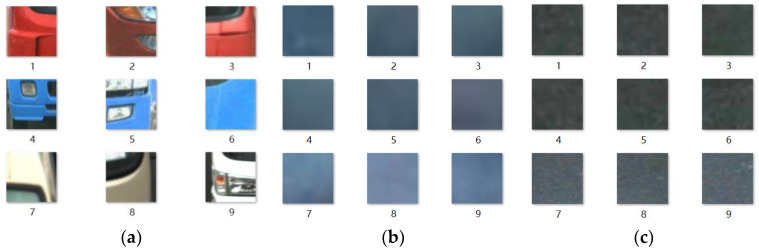
Training samples obtained by the SLIC: (**a**) motor vehicles’ black smoke; (**b**) black smoke emissions; (**c**) motion shadows.

**Figure 9 sensors-23-08281-f009:**
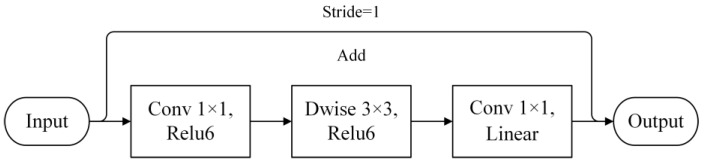
Inverted residual module composition structure.

**Figure 10 sensors-23-08281-f010:**
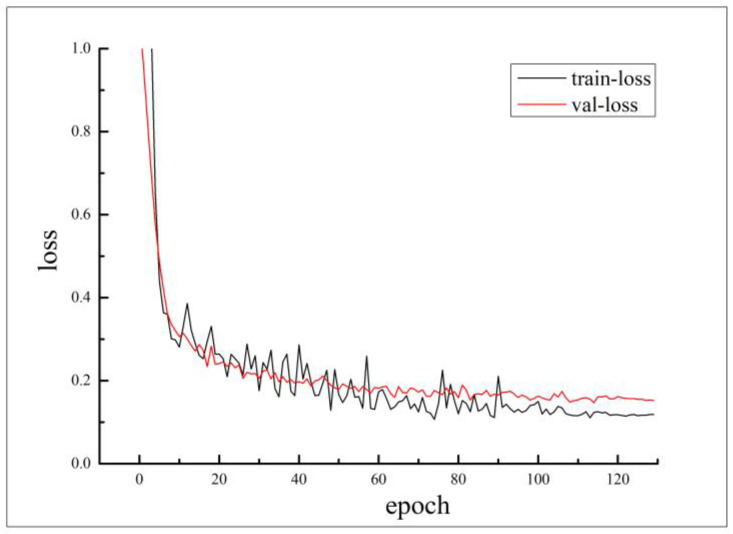
Loss function variation curve of Y-MobileNetv3 model.

**Figure 11 sensors-23-08281-f011:**
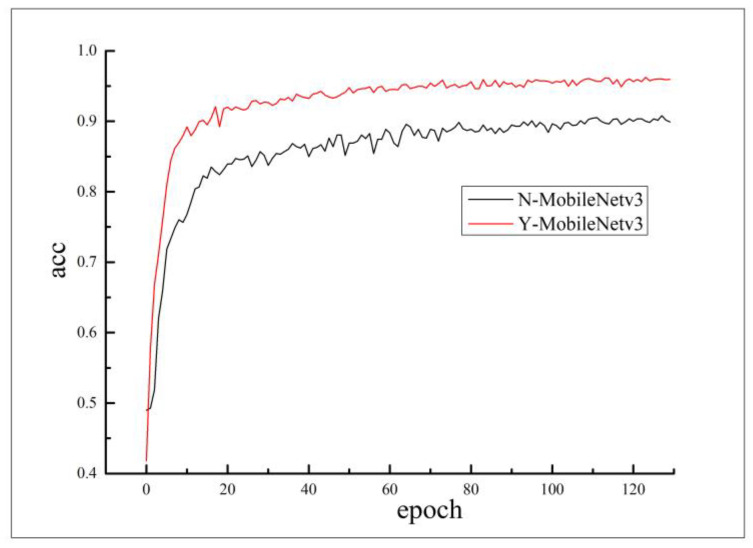
Average accuracy change curve based on MobileNetv3 models.

**Figure 12 sensors-23-08281-f012:**
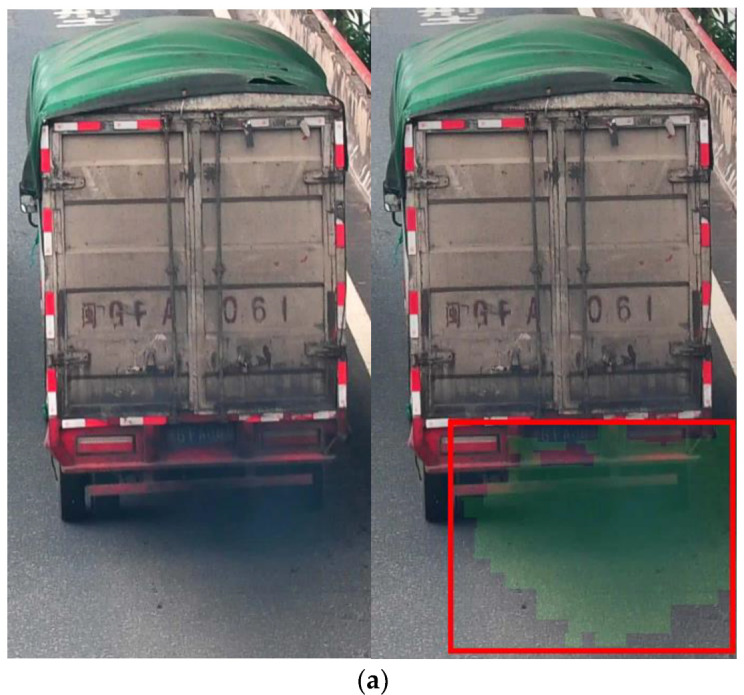
Test results of the Y-MobileNetv3 model: (**a**) black smoke exhaust; (**b**) motion shadows; and (**c**) coexistence of both.

**Figure 13 sensors-23-08281-f013:**
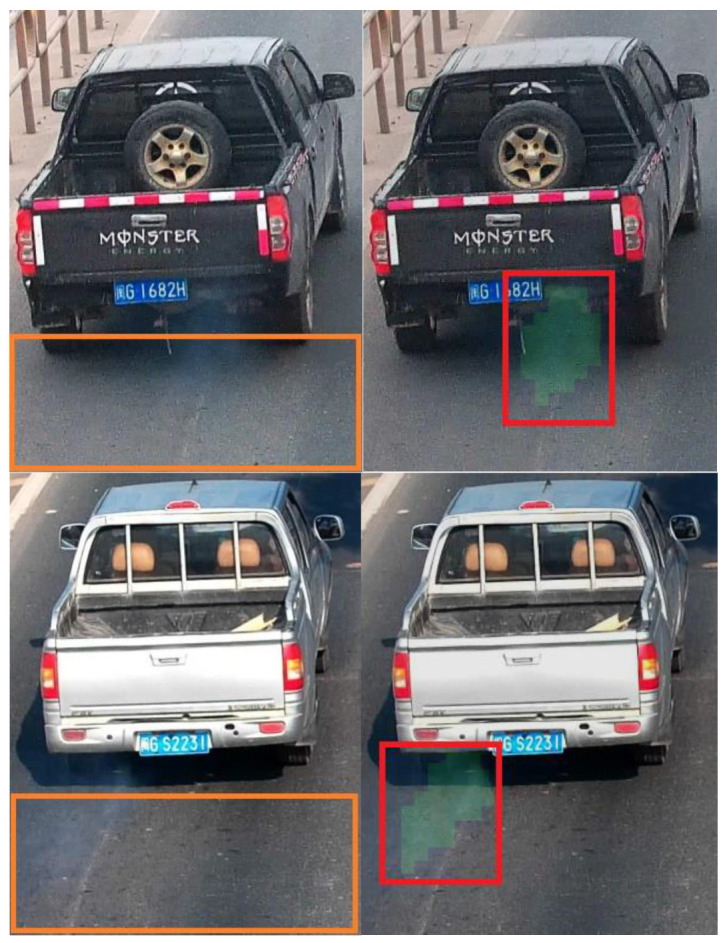
The N-MobileNetv3 (**left**) and Y-MobileNetv3 (**right**) models recognize motor vehicles emitting trace amounts of black smoke exhaust.

**Table 1 sensors-23-08281-t001:** Comparison of different models in the YOLO series.

Model	Batch_Size	mAP (%)	Params (M)	FLOPs (G)
YOLOv5s	256	95.8	7.2	16.5
YOLOXs	128	95.2	9.0	26.8
YOLOv8s	128	94.5	11.2	28.6

**Table 2 sensors-23-08281-t002:** Experimental environment configuration.

Name	Version Model
Operating system	Windows 10
CPU	Intel(R) Core (TM) i5-11400F @2.60 GHz
GPU	NVIDIA GeForce GTX 1650
Programming language	Python 3.8.13
Deep learning framework	Pytorch 1.13.0, CUDA 11.7

**Table 3 sensors-23-08281-t003:** Test results based on MobileNetv3 modeling.

Confusion Matrix		Y-MobileNetv3		N-MobileNetv3	
		Smoke	No Smoke	Smoke	No Smoke
True value	smoke	145	8	138	15
	no smoke	6	131	13	124

**Table 4 sensors-23-08281-t004:** Evaluation metrics based on MobileNetv3 model.

Motion Shadow	Average Accuracy	Precision	Recall	Inference Speed per Image
N-MobileNetv3	90.34%	91.39%	90.19%	8.7 ms
Y-MobileNetv3	95.17%	96.03%	94.77%	7.3 ms

**Table 5 sensors-23-08281-t005:** Test results of different algorithms for detecting black smoke vehicles.

Model	P (%)	mAP (%)	FPS (ms)
Improved LeNet-5 [3]	87.34	86.75	8.2 ms
CenterNet-ResNet18 [2]	89.67	90.54	22.2 ms
2D-3D Fusion Network [4]	88.93	87.45	45.9 ms
YOLOv3-M3-CBAM [5]	92.57	93.80	49.2 ms
Ringelman-3D CNN [6]	88.56	86.74	5.9 ms
Ours	96.03	95.17	7.3 ms

## Data Availability

The data presented in this study are not publicly available due to privacy restrictions.

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
