# Peer review of "Automatic Detection Method for Black Smoke Vehicles Considering Motion Shadows"

_sensors, 2023, doi:10.3390/s23198281_

Round 1

Reviewer 1 Report

The manuscript presents an interesting study on “Automatic Detection Method for Black Smoke Vehicles Considering Motion Shadows”. While the research has good practical significance, there are some areas that require attention and revision to improve the clarity, organization, and depth of the paper. I suggest to address the following issues:

1. It is essential to clearly define and emphasize the contributions made by this research. The paper lacks a precise statement, regarding the specific advancements or novel aspects introduced. To enhance the quality and the impact of the paper, I recommend providing a more explicit description of the contributions, outlining how they address existing gaps in the automatic detection of black smoke vehicles (The outline of contributions also needs to be put in the introduction section).

2. There lacks an overall framework/procedure of the proposed method. I have difficulty to understand the entire process of smoke detection of motion vehicles. A comprehensive description of the methodology (including data preprocessing, networks, input & output of each module, final output, etc.), is needed. Clear visualization, like a diagram or a flowchart, would enhance the presentation. These revisions are essential to ensure the classification part is well-documented, transparent, and easily understandable.

3. The authors have adopted to employ YOLOV5s for the task of locating and extracting regions of moving vehicles, laying the groundwork for subsequent image segmentation and classification. However, it is imperative to provide a clear justification for the selection of YOLOV5s, over the recent models, such as YOLOV8, which has demonstrated improved localization accuracy. Therefore, the authors can provide transparency regarding their choice of YOLOV5 and enhance the credibility and comprehensibility of their research within the context of model selection and application suitability.

4. The construction of the dataset is not clearly described. The presentation of the dataset in the manuscript requires reorganization to enhance clarity and facilitate a better understanding for readers.

1) Regarding the use of a superpixel segmentation algorithm to create dataset, how does it exactly help to make the dataset?  The paper should comprehensively outline essential information such as how the dataset was labeled and assigned, annotation details (e.g., 0-smoke, 1-shadow, 2-other), and any specific characteristics such as the sample resolution. 

2) There are some confusing information. For example: (1) Figure 5 presentation is not so clear, and it is difficult to separate (a) and (b). (2) How did a total of 2900 images get subjected to superpixel segmentation resulting in 1082 images of black smoke emissions, 1035 images of motion shadows, and 1118 images of vehicles as training samples for the Y-MobileNetv3 model?

3) While the application of superpixel segmentation to process images of moving objects extracted by the YOLOV5s model is a valid approach for simplifying the input data and addressing challenges like dynamic shadows, it's important to acknowledge potential drawbacks. Superpixel segmentation can indeed introduce complexities related to over-segmentation, under-segmentation, irregular shapes, and parameter tuning, which may affect object detection accuracy. Therefore, I suggest that the authors provide additional details on how they mitigate these challenges and evaluate the impact of superpixel segmentation on their specific detection task. Additionally, discussing potential alternatives or improvements to address these issues would strengthen the paper's contribution and its relevance to the field of vehicle emission detection.

5. The MobileNetV3 classification section needs significant improvement. It lacks crucial details and differentiation between 'Y-MobileNetV3' and 'N-MobileNetV3' models. 

6. In the results section, this paper only presents their own results. A comparison with other methods should be included to verify the effectiveness of their own method. Moreover, a deeper discussion on the significance and implications of those results should be provided. 

7. Figure 10 showcases the impressive results of the "Y-MobileNetV3" model in accurately identifying motor vehicles emitting trace amounts of black exhaust smoke. To provide a more comprehensive understanding and enable readers to make visual comparisons, it would be highly valuable to include the results of the "N-MobileNetV3" model alongside. This addition would enhance the presentation and facilitate a direct visual comparison between the two models, further strengthening the paper's contribution.

8. The symbols in the formulas must be explained.

There are not many grammartical errors. But, academic writing requires a clear presentation of the proposed method, formulas, and results.

Reviewer 2 Report

It is an interesting work, and the manuscript is well organized. However, some issues still should be addressed.

1.       The abstract should generally include the research background and purpose(i.e., what is the research gap?), research methods, research results(preferably with some numerical results), research importance and potential impact. The number of words should be controlled to about 200-400. It should be modified to show the academic contribution and achievement of the manuscript more clearly.

2.       Some questions need to be clarified in the abstract, for example, why should smoky vehicles be monitored?

3.       The author's introduction needs to be optimized, and we suggest that the author evaluate what needs to be improved in the introduction according to the following criteria.

         What is the problem to be solved?

         Are there any existing solutions?

         Which is the best?

         What is the main limitation of the best and existing approaches?

         What do you hope to change or propose to make it better?

         How is the paper structured?

4.       In what ways does this paper contribute to the academic community? Improvements to the model or something else? Please make it clear.

5.        Several new publications about target detection and reconstruction could not be ignored, which should be added in the revised form such as: https://doi.org/10.3390/land12091813; https://doi.org/10.3390/electronics12183799;  10.1109/TGRS.2023.3263848; 10.1109/TCI.2023.3241547

6.       How to define the optimal segmentation result, and why is K=1500?

7.       “The dataset used in this research is derived from road traffic surveillance videos”, where can the readers find that dataset? Or where is the dataset from?

The English of your manuscript should be improved before resubmission. We strongly suggest that you obtain assistance from a colleague who is well-versed in English or whose native language is English.

Reviewer 3 Report

The paper reports a 4.73% improvement in recognition accuracy when considering motion shadows, which is promising. However, it would be helpful to know the statistical significance of this improvement. Were any statistical tests conducted to validate that this improvement is not due to random variations in the data? Additionally, how does this performance compare with existing state-of-the-art methods?

The paper successfully addresses the computational efficiency with an average inference time of 7.3ms per image. However, the paper could benefit from a discussion on the real-world applicability of the proposed method. Are there specific conditions under which the model's performance may degrade? For example, how well does the model handle different lighting conditions, weather, or types of vehicles? A section discussing the limitations and potential areas for future work would make the research more robust. 

To further enrich the context and robustness of your work, it would be advisable to refer to the following two papers that deal with vehicle detection and image transformation in autonomous driving systems:

  1. Ding, F., Mi, G., Tong, E., Zhang, N., Bao, J., Zhang, G., 2022. "Multi-channel high-resolution network and attention mechanism fusion for vehicle detection model." Journal of Automotive Safety and Energy, 13 (1): 122-130. DOI: 10.3969/j.issn.1674-8484.2022.01.012.

  2. Dong, J., Chen, S., Miralinaghi, M., Chen, T., Labi, S., 2022. "Development and testing of an image transformer for explainable autonomous driving systems." Journal of Intelligent and Connected Vehicles, 5(3), 235-249. DOI: 10.1108/JICV-07-2022-0021.

These references could offer additional perspectives and methodologies that might be beneficial for your study, and they could also help readers understand how your work fits into the broader research landscape.

-

Round 2

Reviewer 2 Report

The manuscript has been improved a lot according to the reviewers' comments. The authors carefully checked the whole manuscript and addressed all the comments seriously. 

Reviewer 3 Report

-

-